# A secreted protease-like protein in *Zymoseptoria tritici* is responsible for avirulence on *Stb9* resistance gene in wheat

Reda Amezrou[1]*, Colette Audéon[1], Jérôme Compain[2], Sandrine Gélisse[1], Aurélie Ducasse[1], Cyrille Saintenac[3], Nicolas Lapalu[1,2], Clémentine Louet[1], Simon Orford[4], Daniel Croll[5], Joëlle Amselem[2], Sabine Fillinger[1], Thierry C. Marcel[1]*

1 Université Paris-Saclay, INRAE, UR BIOGER, Palaiseau, France, 2 Université Paris-Saclay, INRAE, UR URGI, Versailles, France, 3 UCA, INRAE, GDEC, Clermont-Ferrand, France, 4 Crop Genetics, John Innes Centre, Norwich, United Kingdom, 5 University of Neuchâtel, Laboratory of Evolutionary Genetics, Neuchâtel, Switzerland

* reda.amezrou@inrae.fr (RA); thierry.marcel@inrae.fr (TCM)

## Abstract

*Zymoseptoria tritici* is the fungal pathogen responsible for Septoria tritici blotch on wheat. Disease outcome in this pathosystem is partly determined by isolate-specific resistance, where wheat resistance genes recognize specific fungal factors triggering an immune response. Despite the large number of known wheat resistance genes, fungal molecular determinants involved in such cultivar-specific resistance remain largely unknown. We identified the avirulence factor *AvrStb9* using association mapping and functional validation approaches. Pathotyping *AvrStb9* transgenic strains on *Stb9* cultivars, near isogenic lines and wheat mapping populations, showed that *AvrStb9* interacts with *Stb9* resistance gene, triggering an immune response. *AvrStb9* encodes an unusually large avirulence gene with a predicted secretion signal and a protease domain. It belongs to a S41 protease family conserved across different filamentous fungi in the Ascomycota class and may constitute a core effector. *AvrStb9* is also conserved among a global *Z. tritici* population and carries multiple amino acid substitutions caused by strong positive diversifying selection. These results demonstrate the contribution of an 'atypical' conserved effector protein to fungal avirulence and the role of sequence diversification in the escape of host recognition, adding to our understanding of host-pathogen interactions and the evolutionary processes underlying pathogen adaptation.

**Data Availability Statement:** Genome sequencing data for the 103 French Z. tritici isolates is available at NCBI BioProject PRJNA777581 (accession

## Author summary

Fungal avirulence (*Avr*) genes are involved in gene-for-gene relationships with host resistance genes. *Avr* genes may at the same time target host defenses to allow infection and be recognized by a host resistance gene triggering a defense response. The fungus *Zymoseptoria tritici* causes Septoria tritici blotch, a major disease of wheat worldwide. *Z. tritici* populations rapidly adapt to selection pressures such as host resistance, leading to

numbers SRR16762555 to SRR16762657). AvrStb9 gene sequences are available at NCBI Genbank (accession numbers OP376566 and OP376567).

**Funding:** T.C.M. received support from the French Fund to support Plant Breeding (FSOV 2008 H). J. A. and T.C.M. received support from the French National Research Agency (program BIOADAPT, grant no. ANR-12-ADAP-0009-04), including the salary of J.C. S. F. and T.C.M received support from BASF France SAS (V.I.S.A. project), including the salary of R.A. INRAE BIOGER benefits from the support of Saclay Plant Sciences-SPS (ANR-17-EUR-0007). The funders had no role in study design, data collection and analysis, decision to publish, or preparation of the manuscript.

**Competing interests:** The authors have declared that no competing interests exist.

resistance breakdown. We report the identification of the avirulence gene *AvrStb9* based on genetic mapping, sequence polymorphisms and allele swapping. *AvrStb9* is involved in the interaction with *Stb9* resistance gene following the gene-for-gene model, and its recognition hinders disease symptoms in hosts carrying the corresponding resistance gene. Unlike other known *Z. tritici Avr* effectors, *AvrStb9* encodes for an unusually large Avr protein with a predicted protease S41 domain conserved among diverse ascomycete lineages. We also highlight several gene mutations likely involved in escaping *Stb9*-mediated recognition.

## Introduction

Filamentous fungal pathogens are responsible for some of the most damaging crop diseases [1]. Their ability to cause disease is facilitated by fungal proteins called effectors, which are delivered into the plant apoplast or cytoplasm during the infection process [2,3]. Effectors play critical roles in the infection by manipulating plant immune system and other cellular functions. To counteract fungal infections, plants have evolved efficient defense systems relying on the recognition of different fungal factors [4]. The first type of defense involves plant receptors recognizing generic fungal molecules, called pathogen-associated molecular patterns (PAMPs). This PAMP-triggered immunity (PTI) provides basal defense against non-adapted pathogens. The second type of defense is activated with the recognition of fungal effectors by plant receptors, defined as resistance (*R*) genes. Effector-triggered immunity (ETI) induces a specific and strong immune response thought to limit the infection [4]. This interaction typically follows the gene-for-gene model, where host *R* gene products recognize pathogen effectors, which are then called avirulence (*Avr*) genes [5]. However, ETI exerts a strong evolutionary pressure on pathogen populations leading to the selection of virulent isolates overcoming host immune response [2]. Most of these virulent isolates displayed mutations in the avirulence gene leading to variant proteins not recognized by the corresponding R protein [6]. These observations emphasize the importance of pathogen *Avr* genes surveillance and the adequate spatiotemporal deployment of *R* genes in the field.

Most of the characterized fungal effector genes encode secreted proteins and lack sequence similarities with other species [7,8]. However, the analysis of protein structures suggests that fungal effectors can be grouped in structurally related families [9–11]. In addition, a number of effectors in plant pathogenic fungi are predicted to encode proteins that share similarities with known enzymes or enzyme inhibitors (reviewed in [8]), *e.g.* some *Avr* genes from *Cladosporium fulvum* (*Avr2* and *Avr4*), *Magnaporthe oryzae* (*Avr-Pita*) or *Melampsora lini* (*AvrM14*). However, only few have been reported to be involved in fungal avirulence, and their identification will broaden our knowledge on the complex molecular cross-talk between plants and pathogens.

*Zymoseptoria tritici* is an ascomycete fungal pathogen that causes the major wheat foliar disease Septoria tritici blotch (STB). *Z. tritici* is apoplastic with a biphasic infection process [12]. After penetration into wheat leaves through stomata, the fungus colonizes the apoplast for 8–11 days while the host remains symptomless [13]. This long asymptomatic phase is followed by a rapid onset of necrotic symptoms, associated with the development of fungal asexual fruiting bodies called pycnidia, which contain the pycnidiospores that further spread the disease [14]. STB is currently the economically most important wheat disease in Europe and is mainly controlled using fungicides and genetic resistances [12,15]. However, this disease remains a problem due to the ability of fungal populations to overcome control measures (*i.e.*, fungicide

resistance, host resistance breakdown) and adapt to climate changes [15]. Despite its economic importance, the molecular basis of *Z. tritici* avirulence remains largely unknown. Only two avirulence genes have been characterized so far, *AvrStb6*, recognized by the resistance protein *Stb6* leading to a strong immune response; and *Avr3D1* which is thought to trigger quantitative resistance in cultivars carrying *Stb7* resistance gene [16,17,18]. Both genes encode small secreted proteins located in highly plastic genomic regions rich in transposable elements thought to favor their rapid diversification [16,18]. In addition to *Stb6* and *Stb7*, 21 major isolate-specific STB resistance genes have been mapped in wheat, but their corresponding avirulence genes remain unknown [19,20,21].

Here, we aimed to broaden our understanding of avirulence in fungal pathogens, the mechanisms underlying gene-for-gene interactions in the *Z. tritici*–wheat pathosystem and the evolutionary processes related to the escape of host recognition. We first showed that a large protein encoding a predicted secreted protease, *AvrStb9*, is responsible for avirulence in a subset of fungal strains. We demonstrated that recognition of *AvrStb9* by hosts carrying *Stb9* resistance gene triggers an immune response following the gene-for-gene model. We next showed that *AvrStb9* belongs to a family of proteins with a predicted protease S41 domain conserved across diverse ascomycete lineages and that mutations in *AvrStb9*, including some in the predicted protease domain, led to variants escaping *Stb9*-mediated recognition.

## Results

### Mapping of a candidate gene conferring avirulence on the cultivar 'Soissons'

'Soissons' is a French bread wheat variety, developed by Florimond Desprez and released in 1987, which represented 10 to 40% of the French wheat acreage between 1991 and 2002. 'Soissons' is moderately susceptible to STB [22] but it has been postulated to carry the resistance gene *Stb9* based on screening with differentially virulent isolates. Infection assays of the fungal population showed that ~12% of the French isolates are avirulent on cultivar 'Soissons' (S1 Fig). We conducted GWAS to identify the *Avr* gene interacting with the resistance in 'Soissons'.

A total of 103 *Z. tritici* strains were re-sequenced, generating 718,810 SNPs passing quality filters. Using full SNP data, we investigated the population structure and kinship and found very little genetic substructure and relatedness in the fungal population (S2 Fig). For instance, the first three PCs accounted for only 3.11% of the total genetic variation. Using a linear mixed model to associate genetic variations with pycnidia (PLACP) and necrosis (PLACN) variations on 'Soissons', we identified a cluster of 26 SNPs on chromosome 1 from 2,092,563 bp to 2,096,143 bp, significantly associated with PLACN at the Bonferroni threshold (Fig 1A). The same SNP cluster was also associated with variation in PLACP but failed to reach the genome-wide significance (S3 Fig). We also detected a significant association with PLACN variations ($p$ = 3.801e-8; position 2,451,010 bp) on a segment of chromosome 7. However, the region falls in a broad section of the chromosome without any *in planta* expressed genes [23]. The significant SNPs on chromosome 1 mapped to a region containing two predicted genes, *Zt_1_692* and *Zt_1_693*. The lead SNP ($p$ = 7.31e-10; position 2,095,552 bp) mapped to the fourth exon of *Zt_1_693*, but was in strong linkage disequilibrium with other variants spanned across the two predicted genes (Fig 1B). *Zt_1_692* is not expressed *in planta* so we were unable to use RNA-seq data to evaluate the quality of the gene annotation and blastp searches yielded only weak hits. Without any corroborating evidence for a biological role, we concluded that *Zt_1_692* is likely a pseudogene or an annotation error. In contrast, *Zt_1_693* is highly expressed *in planta* (S4 Fig), with a peak expression at 9 days post infection coinciding with

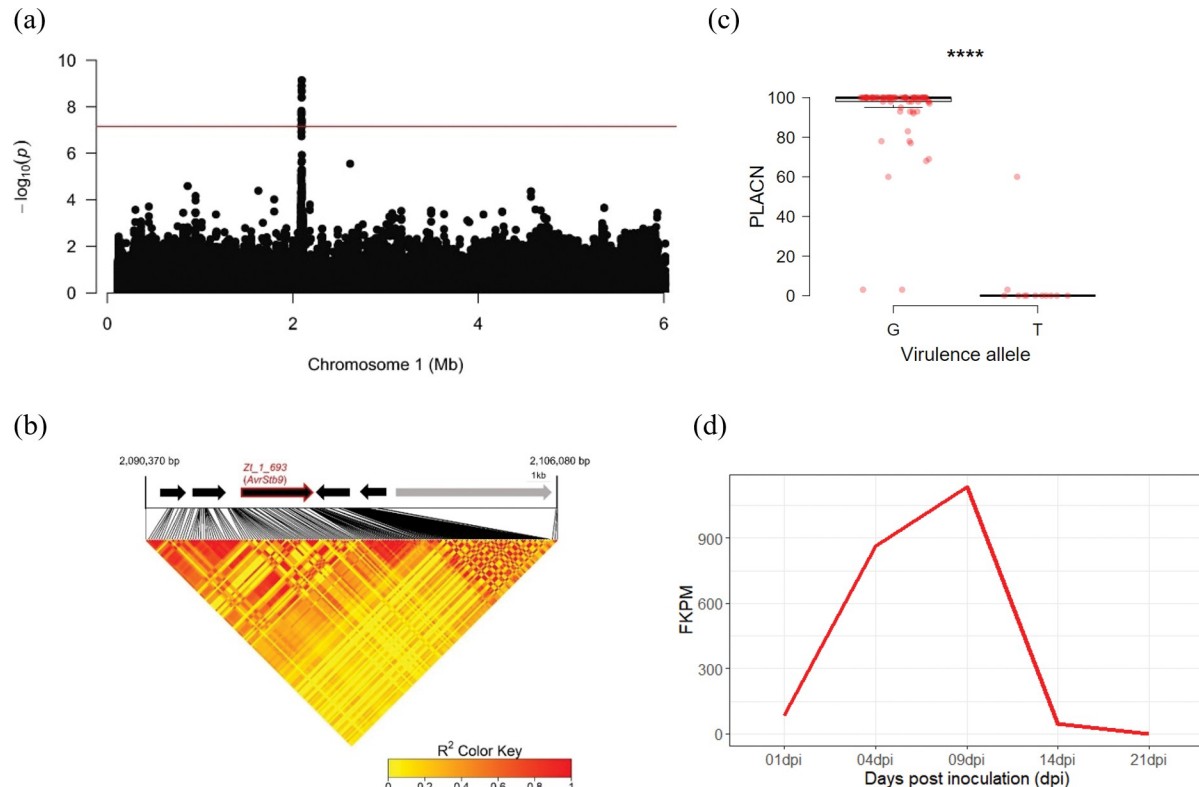

**Fig 1. GWAS identifies a candidate gene controlling avirulence on the cultivar 'Soissons'.** (a) Manhattan plot representing the association of multiple SNPs with the percentage of leaf area covered by necrosis (PLACN) on cultivar 'Soissons'. The X and Y axes indicate positions on chromosome 1 and −log10 (*p*-value) for associations, while the red line represents the Bonferroni threshold (α = 0.05). (b) Top: Gene annotations (black arrows) and transposable elements (grey arrow) on the reference genome IPO323. The candidate gene is highlighted in red. Bottom: Pairwise linkage disequilibrium plot around the lead significant SNP. The association peak is associated with several polymorphisms in the gene *Zt_1_693*. (c) The difference in PLACN of isolates carrying the virulent and avirulent alleles (based on the lead SNP). Significance difference from the mean was computed using a two-sided Mann-Whitney test (****, *p*<0.0001). (d) Gene expression profile of *Zt_1_693* measured as fragments per kilobase of exon per million mapped fragments (FPKM) throughout the time course of *Z. tritici* infection on a susceptible wheat cultivar [23].

the switch from asymptomatic to necrotrophic phase (Fig 1D). Further, allele analyses of the lead SNP clearly discriminated between avirulent and virulent phenotypes (Fig 1C). Taken together, we considered *Zt_1_693* as the best candidate gene to explain avirulence on the wheat cultivar 'Soissons'.

## *Zt_1_693* is involved in a gene-for-gene relationship with *Stb9* resistance gene

To determine whether *Zt_1_693* is an avirulence factor recognized by an *R* gene in the cultivar 'Soissons', we sought to generate *Zt_1_693* deletion mutants (KO) in the virulent strain IPO323 and four different avirulent strains by targeted gene replacement. We also switched alleles between avirulent (IPO09395) and virulent (IPO323) strains using targeted gene replacement. However, we could not obtain any deletion mutant neither in IPO323 (144 transformants tested), nor in the avirulent strains IPO09593, IPO09359 and IPO09139 (n = 95). Only a single deletion mutant was obtained in strain SYN32 out of 32 transformants, producing small colonies and weak symptoms on the susceptible cultivar 'Taichung-29' (S1 Table). This suggests that *Zt_1_693* could be essential for *Z. tritici* growth, and pathogenicity.

Nevertheless, using allele swapping we were able to generate mutants expressing the virulent and avirulent versions of *Zt_1_693* in IPO323 genetic background, and performed infection assays on cultivars 'Soissons', 'Courtot' which carries the *Stb9* gene and a 'Courtot' near isogenic line lacking *Stb9*.

IPO323 gene replacement transformants carrying the avirulent allele (n = 6) from IPO09593 (referred to as *Avrstb9*) were avirulent on 'Soissons' and 'Courtot', but not on 'Courtot' lacking *Stb9* (Fig 2A). Inversely, gene replacement transformants with the virulent allele (referred to as *avrstb9*) from IPO323 (n = 2) resulted in virulence on all tested cultivars, with PLACP values at 26 dpi ranging from 32 to 100% (Fig 2A). Interestingly, Sanger sequencing of IPO323$_{Avrstb9}$ transformants revealed that four strains were chimeric with the avirulent allele not being introduced in full length, retaining some polymorphisms of the virulent haplotype between residues 276 and 628. All four chimeric strains were virulent on *Stb9*-containing cultivars, highlighting the importance of these polymorphisms in virulence (Tables 1 and S1; S5 Fig). The fact that all IPO323 mutants carrying the complete *AvrStb9* allele were unable to infect *Stb9*-containing cultivars, while mutants expressing the *avrStb9* allele produced symptoms on all three cultivars, suggest that *Zt_1_693* encodes the avirulence factor *AvrStb9*, which triggers complete resistance in cultivars carrying the cognate resistance gene.

To further validate the *AvrStb9-Stb9* gene-for-gene relationship, we performed two independent genetic analyses using a bi-parental mapping population and a panel of wheat varieties. The isolate-specific resistance to IPO09593 (*AvrStb9*) was mapped by a QTL analysis on Beaver/Soissons recombinant double haploid progenies. A single major resistance QTL was detected on the long arm of the wheat chromosome 2B, likely corresponding to the *Stb9* locus [24]. This QTL had a log-likelihood (LOD) of 10.77 and 15.33 for the RAUDPC of necrosis and sporulation, respectively (Fig 2B). As expected, the resistance allele originated from 'Soissons' suggesting that the cultivar indeed carries the *Stb9* resistance. We confirmed this interaction by performing a GWAS analysis on a wheat panel phenotyped using the virulent and avirulent strains. Association mapping results revealed a strong signal associated with RAUDPCs for the avirulent strain IPO09593 (eight SNPs above the Bonferroni-corrected cutoff) in the telomeric region of chromosome 2B, but not with the virulent strain IPO323 (Fig 2C). This confirms that IPO09593 triggers a strong resistance from varieties carrying a resistance QTL in the vicinity of the *Stb9* gene. Additional details on *Stb9*-associated SNPs are provided in the supplemental S2 Table. Finally, based on the lead SNP (AX-89505650; position 808,036,267 bp on chromosome 2B), phenotypic and allele analyses showed that ca. 8.2% of the French varieties likely carry *Stb9* (S6 Fig).

Altogether, our combined functional and genetic analyses demonstrate that *Zt_1_693* is the avirulence gene *AvrStb9* and is involved in a gene-for-gene relationship with *Stb9* resistance in wheat.

## *AvrStb9* encodes a secreted protein with a conserved protease domain shared among diverse ascomycete lineages

Unlike other known fungal effectors, *AvrStb9* encodes a large protein and shares sequence similarities with different fungal species. The re-annotated *AvrStb9* gene model has five introns and encodes a 727 amino acid long protein with eight cysteine residues (Fig 3A and 3B). This protein has a predicted signal peptide sequence of 18 amino acids at its N-terminus and two serine-protease domains (C-terminal processing peptidase S41 family; positions 70–726; Fig 3B) according to InterProScan (Panther entry PTHR-37049, e-value: 2.7 e-64). HHpred searches in protein structure databases identified the secreted protease CPAF as the strongest hit (e-value of 1.5e-27), which is involved in pathogenesis of the human pathogenic bacterium

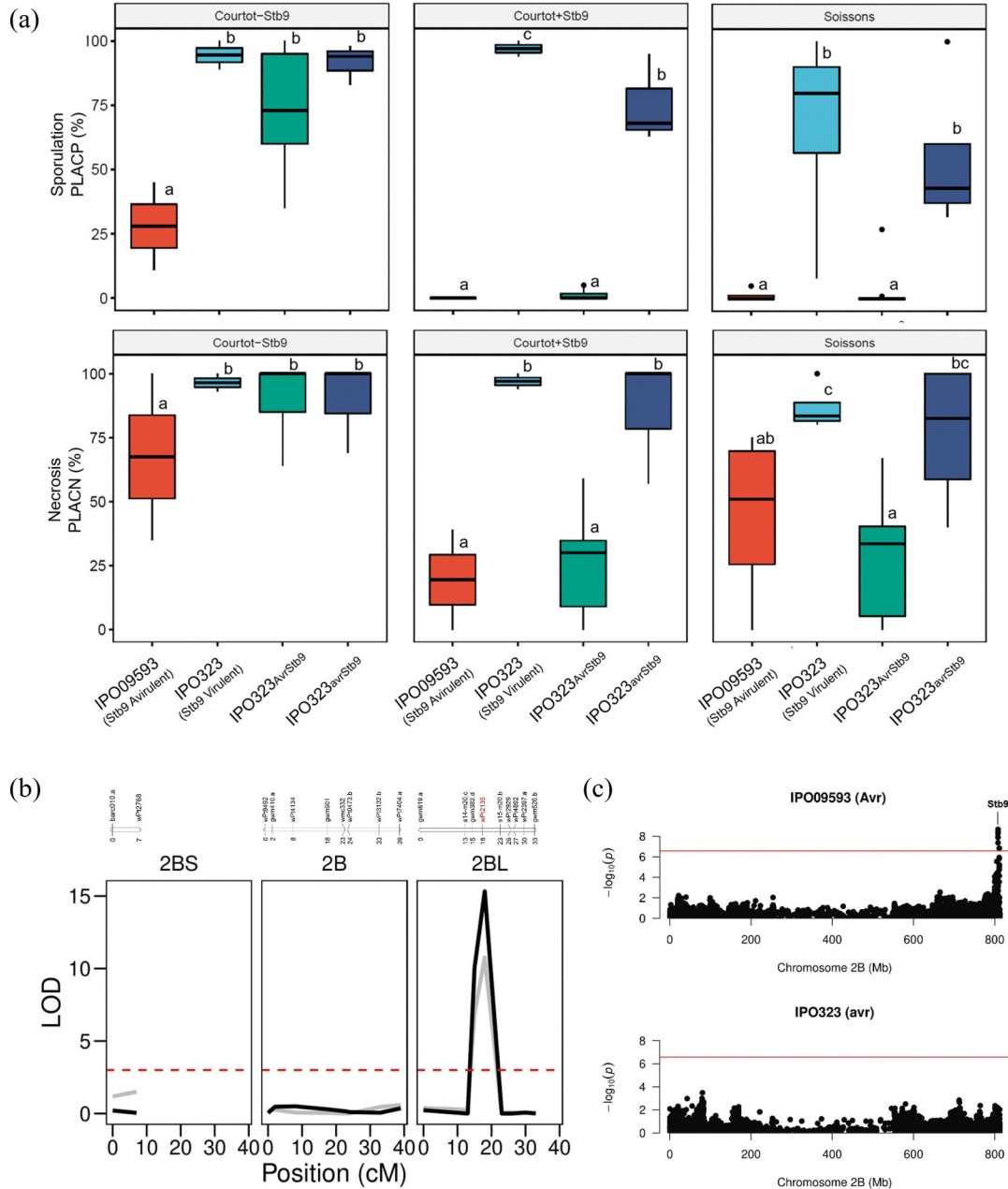

**Fig 2. *Zt_1_693* encodes for the avirulence gene *AvrStb9* and triggers *Stb9* resistance.** (a) Percentage of leaf area covered by pycnidia (PLACP) and percentage of leaf area covered by necrosis (PLACN) at 26 dpi measured after inoculation with the avirulent isolate IPO09593, the virulent isolate IPO323 and IPO323 transformants carrying the avirulent allele (IPO323$_{Avrstb9}$) or the virulent allele (IPO323$_{avrstb9}$) on cultivars 'Courtot' (*Stb9*), a 'Courtot' mutant lacking *Stb9* and 'Soissons' (details on the phenotype of each mutant is provided in S5 Fig; S1 Table). Letter above the boxplots represent statistical groups based on a post-hoc Tukey's HSD test (α = 0.05) performed independently for each wheat background. (b) Genetic mapping of *Stb9* resistance gene using linkage mapping. Top panel: Linkage map and marker position in cM. Bottom panel: Logarithm of odds (LOD) score plot of the QTL analysis (Relative AUDPC for sporulation and necrosis in black and grey, respectively) possibly containing the *Stb9* locus mapped to the long arm of the chromosome 2B. The horizontal dashed line indicates a suggestive linkage threshold (LOD = 3). The position of the marker in the QTL peak is highlighted in red. (c) Manhattan plots representing association mapping results of relative AUDPC sporulation of the avirulent (top) and virulent (bottom) isolates in the bread wheat panel. Multiple SNPs in the telomeric region of chromosome 2B associated with the avirulent isolate are located in the *Stb9* genomic region.

**Table 1. Polymorphisms in AvrStb9 protein sequences between the wild type and transgenic mutant strains.** Difference in amino acid residues between the virulent (avrstb9) and avirulent (Avrstb9) phenotypes are highlighted in orange while residues potentially involved in virulence are highlighted in red.

| Strain | Mutant ID | Codon position / Amino acid residue | | | | | | | | | | | | | Phenotype (on *Stb9*) |
|---|---|---|---|---|---|---|---|---|---|---|---|---|---|---|---|
| | | 58 | 66–71 | 73 | 188 | 276 | 349 | 370 | 431 | 473 | 512 | 525 | 565 | 628 | |
| **Wild-type strains** | | | | | | | | | | | | | | | |
| IPO323 (avrstb9) | - | R | PCNSTI | N | Y | T | A | T | A | F | S | S | D | L | Virulent |
| IPO09593 (Avrstb9) | - | Q | Δ | D | V | N | V | N | D | Y | T | F | N | Y | Avirulent |
| **IPO323 Avrstb9 transformants** | | | | | | | | | | | | | | | |
| IPO323_Avrstb9 hph#5 | E | Q | Δ | D | V | N | V | N | D | Y | T | F | N | Y | Avirulent |
| IPO323_Avrstb9 hph#6 | F | Q | Δ | D | V | N | V | N | D | Y | T | F | N | Y | Avirulent |
| IPO323_Avrstb9 hph#9 | H | Q | Δ | D | V | N | V | N | D | Y | T | F | N | Y | Avirulent |
| IPO323_Avrstb9 hph#12 | K | Q | Δ | D | V | N | V | N | D | Y | T | F | N | Y | Avirulent |
| IPO323_Avrstb9 hph#14 | L | Q | Δ | D | V | N | V | N | D | Y | T | F | N | Y | Avirulent |
| IPO323_Avrstb9 hph#18 | N | Q | Δ | D | V | N | V | N | D | Y | T | F | N | Y | Avirulent |
| **IPO323 Avrstb9 chimeric transformants** | | | | | | | | | | | | | | | |
| IPO323_Avrstb9 hph#3 | I | Q | Δ | D | V | T | A | T | A | F | S | S | D | L | Virulent |
| IPO323_Avrstb9 hph#11 | J | Q | Δ | D | Y | T | A | T | A | F | S | S | D | L | Virulent |
| IPO323_Avrstb9 hph#7 | G | Q | Δ | D | Y | T | A | T | A | F | S | S | D | L | Virulent |
| IPO323_Avrstb9 hph#17 | M | Q | Δ | D | V | N | A | T | A | F | S | S | D | L | Virulent |
| **IPO323 avrstb9 transformants** | | | | | | | | | | | | | | | |
| IPO323_avrstb9 hph#3 | O | R | PCNSTI | N | Y | T | A | T | A | F | S | S | D | L | Virulent |
| IPO323_avrstb9 hph#4 | P | R | PCNSTI | N | Y | T | A | T | A | F | S | S | D | L | Virulent |
| Percentage of virulent isolates in the fungal population | | 87.5 | 98.86 | 86.63 | 98.86 | 98.86 | 54.54 | 88.63 | **98.86** | **98.86** | 89.77 | **98.86** | **98.86** | 95.54 | |
| Probability of dN/dS>1 | | 0.97 | - | 0.99 | 0.99 | 0.58 | 0.54 | 0.59 | 0.93 | 0.60 | **0.99** | **1.00** | 0.65 | **1.00** | |

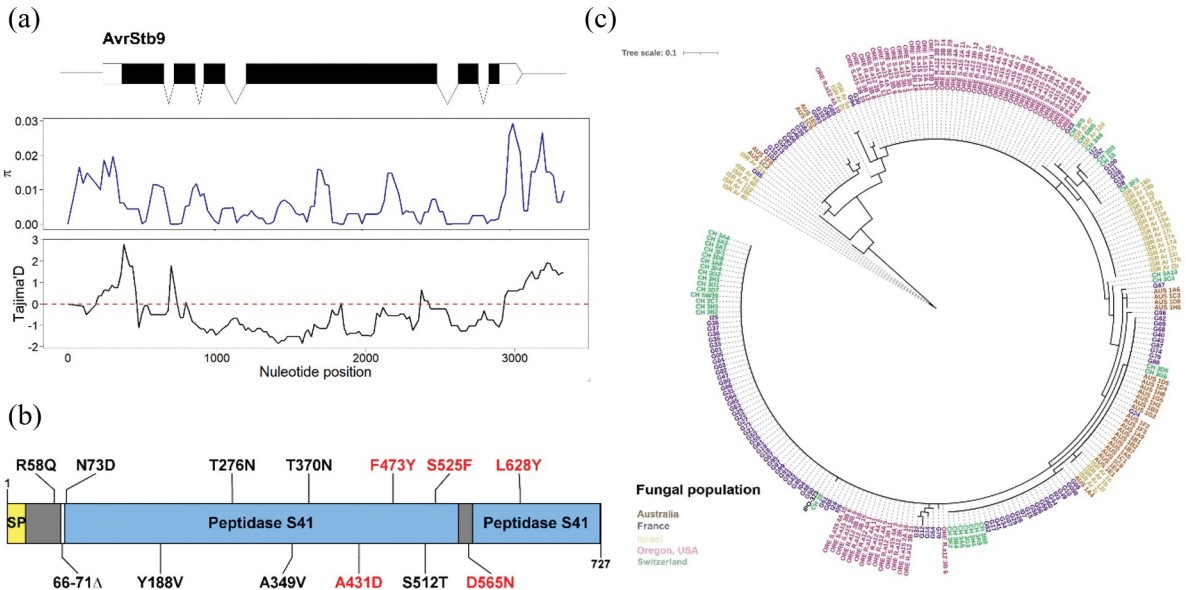

**Fig 3. *AvrStb9* diversity and phylogeny in the global *Z. tritici* population.** (a) The top panel shows the *AvrStb9* gene structure. Middle and low panels are sliding window analyses for nucleotide diversity (π) and Tajima's D across the *AvrStb9* locus, respectively, with a window size of 200 bp and a step size of 20 bp. The plots show a reduced nucleotide diversity and a drop in Tajima's D values, likely corresponding to a selective sweep spanning the region. (b) Schematic diagram of *AvrStb9* protein sequence with an annotation of amino acid substitutions between virulent and avirulent alleles. Substitutions potentially involved in virulence are highlighted in red. SP = Signal peptide. (c) A phylogenetic tree of *AvrStb9* protein sequence generated from a global *Z. tritici* population and the reference isolate IPO323.

*Chlamydia trachomatis* [25]. The predicted protein structure of AvrStb9 was used to search for structural similarity in PDB, which identified the same protease CPAF with highest similarity to AvrStb9 (DALI Z-score 16.5) (S3 Table). Mature CPAF is a homodimer comprised of two distinct subunits, chains A and B [25]. The pairwise structure alignment of AvrStb9 with both chains of CPAF confirms the structural similarities with a TM-score of 0.56 (S4 Table).

To investigate whether *AvrStb9* originated before or after *Z. tritici* speciation, we used blastp searches against *Z. tritci* reference genome IPO323 and closely-related species from the *Zymoseptoria* family. We identified *AvrStb9* and its paralogs from IPO323 and all its closest relatives *Z. pseudotritici*, *Z. ardabiliae* and *Z. brevis* (S7A Fig). Interestingly, most of these proteins had a predicted signal peptide, and they all displayed a C-terminal processing peptidase S41 domain. Using IPO323 *in planta* gene expression, we found that three out of the five paralogs (*Zt_4_489*, *Zt_10_226*, *Zt_11_236*) had an expression profile comparable to *AvrStb9* (S7B Fig). According to these expression patterns, these genes could potentially play a role in infection.

Using OrthoMCL, we identified 122 putative *AvrStb9* orthologs/paralogs with an average identity of 33.6% and an average e-value of 8e-73 (S5 Table). *AvrStb9* orthologs/paralogs are found in a broad range of ascomycete lineages belonging to Eurotiomycetes, Leotiomycetes, Sordariomycetes, Orbiliomycetes and Dothideomycetes (Fig 4; S5 Table). Most of fungal species (48 out of 65 species) identified from the orthogroup are pathogenic, indicating that the well conserved *AvrStb9* orthologs/paralogs may be involved in host interaction processes. Interestingly, ~87% of these proteases possess a predicted signal peptide.

## Diversification of *AvrStb9* leads to evasion of host recognition

Non-synonymous substitutions in avirulence gene sequences often mediate the evasion of host recognition. Analysis of sequence polymorphisms between the avirulent and virulent strains revealed a 6 amino acid deletion and 12 non-synonymous substitutions (Table 1; Fig 3B). We crossed this data with the fungal population phenotypes and identified eight mutations potentially involved in host evasion for which more than ~94% of strains carrying the mutation were virulent on *Stb9* cultivars (Table 1). Five of these mutations fall into the virulence-domain cited above, namely A431, F473, S525, D565, L628. All of these mutations except D565 are located in one of the two putative protease domains (Fig 3B), with F473, S525 and L628 being relatively conserved residues among proteins belonging to the CPAF structural family (in 34–57% of the analyzed proteins respectively). The visualization of these five residues on the predicted protein structure of avrStb9 revealed their physical proximity (Fig 5B). The predicted structures of AvrStb9 and avrStb9 aligned with 94% sequence identity (TM-score of 0.94) (Fig 5A; S4 Table), not indicating any important structural changes between both protein variants. Comparing the physical distances between the five residues on AvrStb9 and avrStb9 predicted structures identified the residue S525 (*vs*. F525) as being much more distant (10–27%) from the other four residues in the virulent avrStb9 compared to the avirulent AvrStb9 predicted structure. In addition, the F to S modification at position 525 replaces a polar residue by a hydrophobic one. This may indicate the importance of the 525 residue in escaping recognition by *Stb9* (Fig 5C and 5D).

In order to investigate *AvrStb9* diversity at a larger scale, we extended our analysis to include four global *Z. tritici* populations. We found 44 different *AvrStb9* alleles in these populations, which translate to 38 different protein variants, mostly population-specific (Fig 3C). Next, we analyzed *AvrStb9* nucleotide diversity including ~500 bp of up- and downstream sequences from the start and stop codons, respectively. We observed both a drop-in nucleotide diversity and Tajima's D in the *Avr* coding sequence compared to the flanking regions (Fig

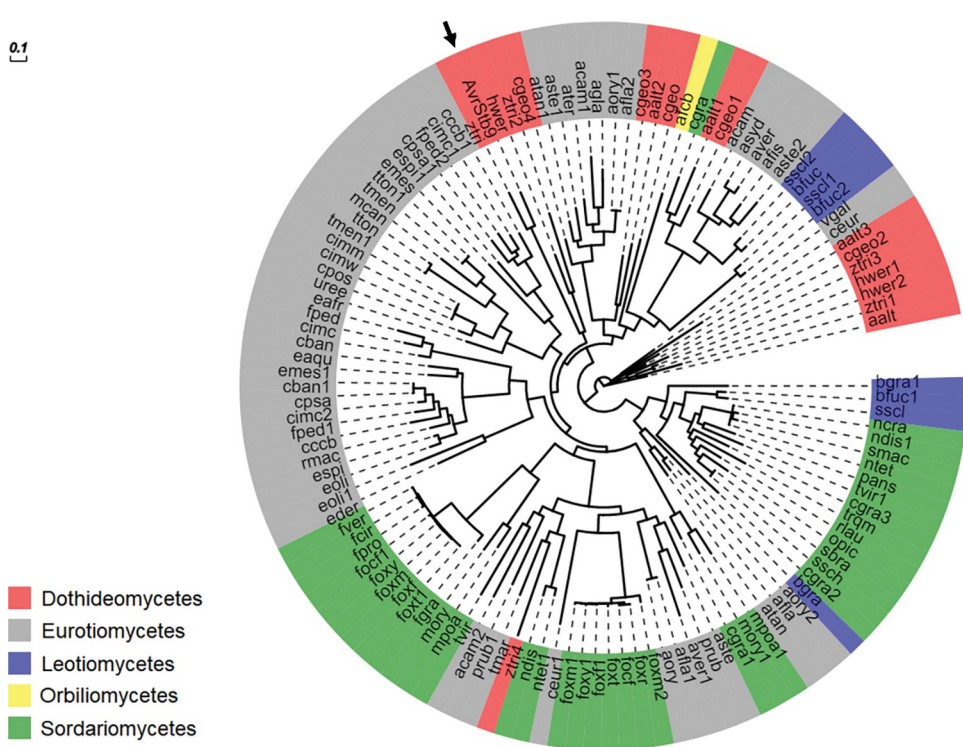

**Fig 4. Phylogenetic tree of selected *AvrStb9* orthologs/paralogs belonging to five distinct classes of Ascomycota lineages.** Leaf labels are color-coded according to class and represent abbreviated names of the fungal species (S5 Table). *AvrStb9* is indicated with a black arrow.

3A). A negative Tajima's D is indicative of an excess of low frequency polymorphisms, indicating population size expansion and/or selective sweeps.

The codon-based model analysis of *AvrStb9* coding sequence was used to investigate the selection pressure exerted on the avirulence gene. Likelihood ratio tests (LRT) for different models indicated that positive selection was highly probable, as selection models (M2 and M8) showed a significantly better data fit compared to the neutral models M1 and M7 (S6 Table). Selection analysis revealed 23 residues (3.16% of the protein sequence) under positive diversifying selection (BEB probability > 0.95) in the M2 model. Interestingly, S525F and L628Y, had the strongest positive selection probability (BEB probability of = 1) among the five mutations potentially responsible of the virulence according to the previous analyses (Table 1). We postulate that these adaptive mutations are under strong selective pressure most probably to counteract recognition by the *Stb9* resistance gene.

## Discussion

Gene-for-gene interactions in plant pathosystems are described when a host resistance gene recognizes a pathogen's avirulence product that triggers a host immune response. Although 23 isolate-specific *Stb* resistance genes have been mapped in wheat, only two have been cloned so far and their corresponding fungal avirulence genes remain largely unknown. Here, we report the identification and characterization of a novel avirulence gene, *AvrStb9*, which triggers complete resistance in wheat cultivars carrying *Stb9* resistance gene. Additionally, we performed a high-resolution mapping of *Stb9* locus and delivered tightly-linked SNP markers for its use in breeding.

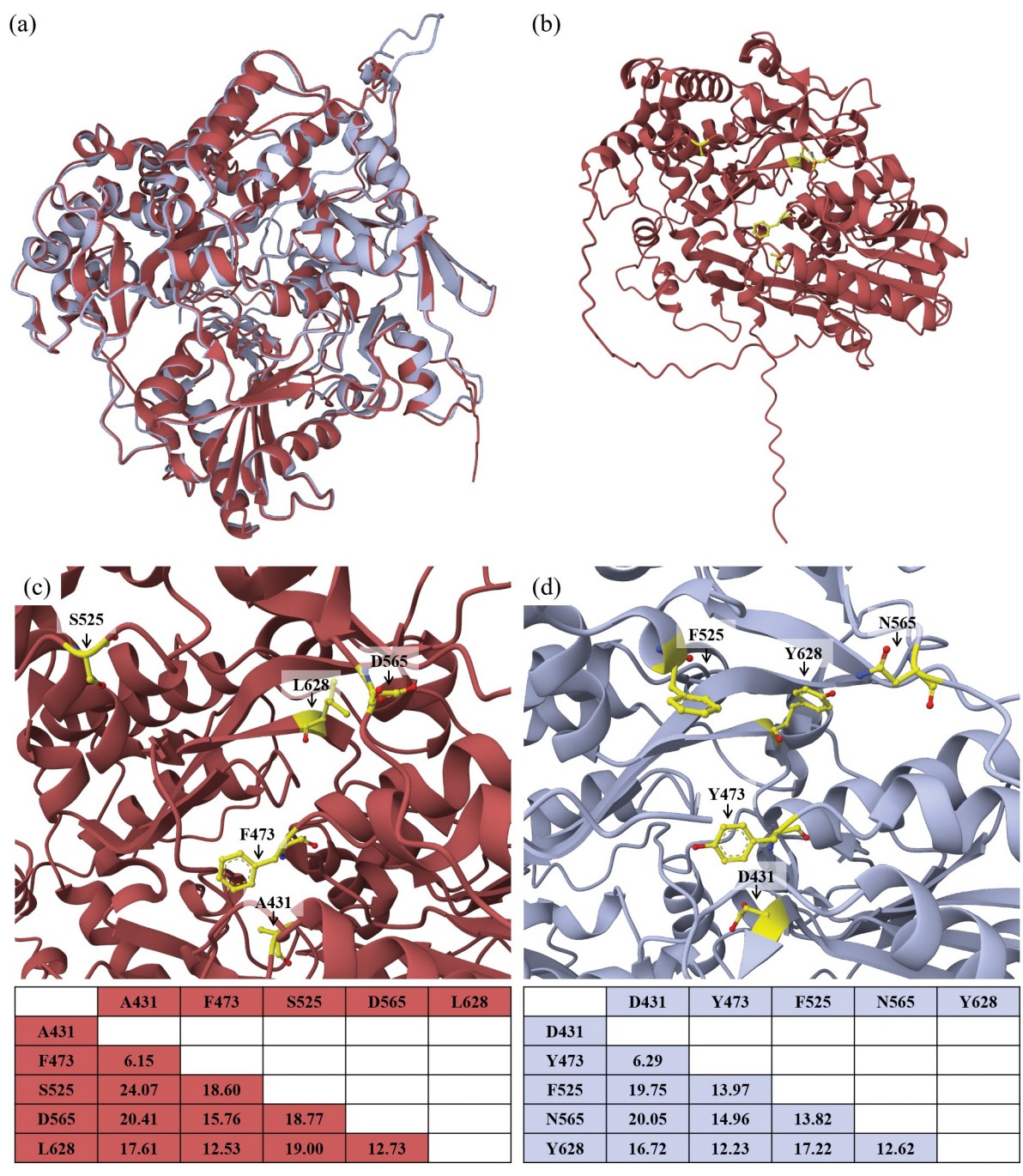

**Fig 5. Three-dimensional structure of the protein AvrStb9 predicted with the ColabFold v1.5.2: AlphaFold2.** (a) An overlay of the 3D structures of the virulent avrStb9 protein (isolate IPO323 in red) and of the avirulent AvrStb9 protein (isolate IPO09593 in blue). (b) The 3D structure of the virulent avrStb9 protein with the localization of the five residues potentially involved in virulence (highlighted in yellow). The (c) virulent avrStb9 and (d) avirulent AvrStb9 proteins with Top panel: zoom-in on the five residues in the 3D structure, and Bottom panel: distances (in Å) between the five residues.

*AvrStb9* is highly expressed during the asymptomatic phase of *Z. tritici* infection and drastically downregulated upon the switch to the necrotrophic phase. In a previous study, *AvrStb9* was among the top 100 highly expressed genes during the transition from biotroph to necrotrophy stage [26], suggesting a function in host colonization during the asymptomatic phase

or the transition to the necrotrophic phase. The late recognition of *AvrStb9* could be indicative that the resistance gene inhibits the transition from biotrophy to necrotrophy (*i.e.* inhibition of cell death). A useful means to verify this hypothesis is through heterologous expression assays, which would help to understand how the *Avr* gene elicits *Stb9*-dependent resistance. Recognition of the *Avr* allele leads to full resistance in cultivars carrying *Stb9* following the gene-for-gene model. A previous study mapped the *Stb9* resistance locus to the long arm of chromosome 2B but the genomic interval remained very large [24]. We used *AvrStb9* strain to trigger wheat resistance and perform a precise mapping of the resistance locus, which revealed that *Stb9* resides in the telomeric region of wheat chromosome 2B in an interval of 79.99–81.10Mb ($r^2 > 0.50$). Only two STB resistance genes have been cloned to date, *Stb6* and *Stb16q*, both encoding receptor-like kinase proteins [27,28], and our findings will help in the cloning of *Stb9* resistance gene.

*AvrStb9* is an atypical fungal avirulence gene, since it encodes a relatively large secreted protein (727 aa). Up to now, most of cloned avirulence genes from filamentous fungal pathogens encode small (< 300 aa), cysteine-rich secreted proteins. Only three exceptions to this rule are the *Ace1* gene from *Magnaporthe grisea* [29] encoding a large polyketide synthase (PKS) fused to a nonribosomal peptide synthetase (NRPS) likely involved in secondary metabolism, and two *AvrMla* genes from *Blumeria graminis hordei* which encode large proteins [30]. Moreover, it appears that AvrStb9 has a predicted protein domain corresponding to C-terminal processing S41 proteases, and show sequence similarities with proteins from diverse fungal species. In plant pathogenic bacteria, some type III secreted effectors injected in host plant cells are similar to catalytic enzymes involved in a variety of reactions such as E3 ligation or proteolysis [31]. For instance, *Pseudomonas syringae* secretes the cysteine protease *HopN1* to suppress the host innate response and promote virulence [32].

The importance of proteases in fungal pathogenicity has previously been reported [33,34]. *Fusarium oxysporum* f. sp. *lycopersicum* secretes a serine protease and a metalloprotease that cleave host chitinases to prevent the degradation of fungal cell walls and facilitate host colonization [35]. The maize anthracnose fungus *Colletotrichum graminicola* secretes a highly conserved metalloprotease effector that enhances virulence [36]. Similarly, the rice blast fungus *Magnaporthe oryaze* secretes during infection *Avr-Pita*, an effector protein with a predicted zinc metalloprotease domain that belongs to a family of proteases that includes other avirulence genes recognized by the resistance gene *Pi-ta* [37,38]. Our protein analyses (OrthoMCL, InterProScan, HHpred) indicated that AvrStb9 has sequence and structural features typical of a protease. Protein structure-based homology searches identified the secreted protease CPAF from the human pathogenic bacterium *C. trachomatis* as closest homolog. CPAF degrades host transcription factors and therefore directly helps *Chlamydia* evading the host defenses. One may speculate that AvrStb9 plays a similar role in proteolytic degradation of host protein substrates. However, the enzymatic activity of AvrStb9 is yet to be investigated, as our analysis relied primarily on sequence and structure comparisons. Further, the fact that we were hardly able to produce *AvrStb9*-disrupted mutants indicates that the gene may play an important role in *Z. tritici* survival, and that the proteolytic activity likely extends to include substrates essential for fungal growth.

The broad conservation of AvrStb9 in *Z. tritici* world population, its sister species and some other fungal species suggested that this protein is a core effector shared among these lineages. Irieda *et al.* [39] reported the conservation of *NIS1* effector that was initially detected from two *Colletotrichum* spp. in a wide range of Ascomycota and Basidiomycota lineages. *NIS1* targets plant immune kinases to suppress PAMP-triggered immunity [39]. We hypothesize that S41 proteases may have been anciently acquired by fungi, and that the expansion of particular serine protease families is associated with the evolution of fungal lifestyle. Indeed, S41 serine

proteases are suspected to have a role in programmed cell death and pathogenic life style adaptation in fungi [40]. We also identified multiple paralogs from multiple fungal species, likely resulting from gene duplication events, highlighting either the redundancy of these genes or their evolution to interact with different substrates. The functional redundancy of virulence genes has been a technical challenge to study their biological functions, as disruption mutants often lead to the conclusion that these genes are not essential for pathogenicity [41,42]. These redundancies may also increase the organism's genetic robustness against null mutations or act in compensation of a gene loss [43]. In *Z. tritici* for instance, the different *AvrStb9* paralogs had a secretion signal and show an expression pattern consistent with a role as effectors. Taken altogether, we hypothesize that *AvrStb9* and related S41 proteases constitute a set of core effectors gained early in the evolution of fungal-plant interactions.

Recognition of effectors by host resistance exerts an evolutionary pressure that favors sequence diversification, effector deletion or acquisition to counteract the immune response. We analyzed AvrStb9 sequence diversity and found a broad conservation of its protein residues, unlike the other known *Z. tritici* avirulence genes, *AvrStb6* and *Avr3D1* [44,18]. Nevertheless, we identified few substitutions undergoing strong positive diversifying selection. Additional data from chimeric AvrStb9 transformants, from *Z. tritici* populations and from sequence conservation among the CPAF family indicate S525 and L628 as probably the most important residues involved in virulence. Moreover, the greater distance and the loss of polarity of residue S525 in the virulent form of avrStb9 may indicate the importance of this residue in escaping recognition by *Stb9*, although we cannot exclude that combinations of mutations may be necessary. Based on the structural model for CPAF function, one may speculate that a functional protease is required to escape host recognition.

This is corroborated by the fact that, we did not observe in any of the global strains a loss of *AvrStb9* or other deleterious mutations. Furthermore, *AvrStb9* deletion proved to be nearly impossible (only one poorly growing transformant had the gene deletion out of the 271 tested). We therefore suggest that *AvrStb9* plays an essential role in the biology of *Z. tritici*. Additional functional studies, including mutating the S41 protease active sites in virulent and avirulent alleles will shed light on the role of the protease activity of AvrStb9 in its diverse biological activities (fungal growth, wheat infection, recognition by *Stb9*). But also, the function of the wheat resistance factor Stb9 is intriguing: Does it work as classical receptor protein, recognizing the avirulent AvrStb9 protein or an AvrStb9-produced peptide? Does it inhibit AvrStb9 protease activity? May the Stb9 resistance protein be the target of AvrStb9 protease, or, on the contrary, does AvrStb9 targets a different protein that is "guarded" by Stb9 resistance?

In conclusion, we identified a new major avirulence factor of *Z. tritici* that triggers an immune response in hosts carrying the corresponding resistance gene. Unlike what has been described for most avirulence genes, *AvrStb9* encodes a large protein with a predicted protease domain and likely belongs to a family of conserved fungal S41 proteases. This is an important step in deciphering molecular mechanisms underlying wheat–*Z. tritici* interactions, and shed light on the role of 'atypical' effectors in fungal avirulence. It opens new research avenues to broaden our knowledge on host-pathogen interactions, in particular the characterization of enzyme activity of effectors, which will greatly help our understanding of the biology of protease effectors, commonly found in fungal pathogens.

## Materials and methods

### Fungal material

A total of 103 *Z. tritici* strains were used for GWAS analysis. The strains derive from a French *Z. tritici* collection collected in 2009 on cultivars 'Apache' (n = 44), 'Premio' (n = 42), 'Soissons'

(n = 9), 'Caphorn' (n = 3), 'Alixan' (n = 1), 'Bermude' (n = 1), 'Dinosaur' (n = 1) and 'Garcia' (n = 1); one isolate was collected in the UK on cultivar 'Humbert'. The two isolates used for genetic mapping of the resistance and fungal transformation are IPO323 and IPO09593. IPO323 is a Dutch isolate collected on cultivar 'Arminda' in 1981 and used as the reference strain for *Z. tritici* [45]. IPO09593 is part of the 103 strains and has been isolated from 'Apache' in 2009. IPO323 is virulent on cultivars carrying *Stb9* resistance gene while IPO09593 is avirulent on *Stb9*-cultivars. The three additional isolates used to construct deletion and allele swap transformants, namely IPO09359 ('Apache', 2009), IPO09139 ('Apache', 2009) and SYN32 (unknown cultivar, 2006), are avirulent on *Stb9*-cultivars.

## Plant material

We performed quantitative trait loci (QTL) mapping in a recombinant doubled haploid (DH) population comprised of 65 lines, developed from the $F_1$ progeny of the cultivars 'Beaver' and 'Soissons' [46]. A panel of 220 modern French bread wheat (BW) varieties was used for GWAS analysis; details on the BW panel used here can be found in [47]. We also characterized *Z. tritici* wild-type and mutant strains on near isogenic lines (NILs) carrying or not the *Stb9* resistance gene in the genetic background of cultivar 'Courtot', developed at INRAE GDEC (Clermont-Ferrand, France). The 'Courtot' NILs were developed following three backcrosses starting with $F_1$ 'Courtot' x 'Chinese spring' (carrying a susceptible allele of *Stb9* gene). To keep the *Stb9* susceptible allele in the progenies, each plant was genotyped with single sequence repeat markers wmc317 and barc129 and phenotyped with the *Stb9* avirulent isolate IPO89011. A $BC_3F_1$ plant heterozygous at the *Stb9* locus was self-fertilized. $BC_3F_2$ plants homozygous either for the susceptible *Stb9* allele ('Courtot'–Stb9) or for the resistant allele ('Courtot' + Stb9) were selfed and constitute the 'Courtot' NILs.

## Pathogenicity assays

Pathogenicity assays were performed following the procedure described by [21]. Briefly, 16 days old plants were marked to delimit a 75 mm segment on the first true leaf. The segment was inoculated with a paintbrush dipped into a $10^6$ spores.mL$^{-1}$ inoculum solution with one drop of Tween20 added per 15 mL. After inoculation, the plants were watered and covered with transparent polyethylene bags for 72 hours to initiate the infection. The plants were kept in a climate chamber with a 16 h photoperiod (300 μmol.m$^{-2}$.s$^{-1}$), 80–90% relative humidity and 18°C night/22°C day. Phenotyping of the fungal population was performed by visually assessing strain virulence on the cultivar 'Soissons' at 21 days post inoculation (dpi). Visual assessments were made of percent of leaf area covered by necrosis (PLACN) and by pycnidia (PLACP) within the inoculated segment of each leaf (disease scale available at S8 Fig). Each of the 103 strains was tested in three replicates with three leaves per replicate. Phenotyping of *AvrStb9* mutant strains was performed following the same procedure.

 For the host mapping populations, the Beaver/Soissons DH lines were inoculated with the strain IPO09593, and the 220 wheat varieties were inoculated with the strains IPO09593 and IPO323. Here, each interaction was tested in two replicates with three leaves per replicate and visual assessments of PLACN and PLACP were recorded at 14, 20 and 26 dpi. We calculated the area under disease progress curve for necrosis (AUDPCn) and sporulation (AUDPCs) with the formula:

$$\text{AUDPC} = \frac{\sum[(t_{i+1} - t_i) \times (y_i + y_{i+1})]}{2}$$

With, $t_{i+1}-t_i$: number of days between two observations; $y_i$: PLACN or PLACP at day i; $y_{i+1}$: PLACN or PLACP at day i+1. AUDPC were calculated relative to their maximum theoretical value, RAUDPCn and RAUDPCs respectively. For all assays, disease severity data were averaged between replicates and used as the input phenotype in GWAS and QTL analyses.

## Fungal whole-genome sequencing and variant calling

Fungal spores were obtained after 7 days growth in liquid cultures of yeast extract-peptone-dextrose Broth (Difco YPD Broth), and then freezed-dried during 24 h. High molecular weight DNA was extracted using a phenol-chloroform based procedure. Illumina paired-end libraries were prepared for each sample and DNA fragments sequenced with 2x100 bp reads on a HiSeq-2000 sequencing system at Genewiz, Inc. (formerly Beckman Coulter Genomics). Depth of sequence coverage was variable depending on the isolate, ranging from 16 to 303 genome equivalents with an average of 61 genome equivalents. For each strain, reads were trimmed using trimmomatic v0.32 [48] and then mapped on the reference genome IPO323 [45] using the mem algorithm from BWA v0.7.7 [49] with default options. Samtools v0.1.19 and Picard tools v1.106 [50] were used to filter reads (optic duplicates, secondary alignments and reads with a mapping quality < 30) and keep only pairs in which both reads met quality checks. SNP calling was done with Freebayes v0.9 with the option—report-monomorphic [51]. All positions detected as low-complexity regions or transposable elements (TE) were excluded using RepeatMasker 4.0 with default options and REPET [52], respectively. Quality filters were applied on the gVCF using in-house scripts. Depth of coverage (DP) calculated at each position follows a normal distribution around the depth main peak; a DP filter was set at twice the standard deviation from the mean. All the gVCF were stored into a matrix reduced to keep only the positions where at least one strain has a detected SNP in a non-filtered position. The SNP matrix was parsed in.tped format, converted to.ped/.map and filtered for < 0.5 missing data and > 0.1 minor allele frequency (MAF) using PLINK v.1.9 [53].

## GWAS analyses

For Genome-Wide Association Analysis (GWAS), we used the linear mixed model implemented in the R package 'GAPIT' [54]. To account for population structure and reduce spurious associations, a kinship matrix using the VanRaden method [55] and the first three principal components computed from SNP data were included a covariate in the model. We used a Bonferroni threshold of $p <$ 2.67e-7 ($\alpha$ = 0.05) to declare significant SNPs associated with phenotypic variations. Linkage Disequilibrium (LD) measures ($r^2$) for the significant SNPs were obtained with PLINK v.1.9 using the—r2 command and LD blocks were visualized using the R package 'LDheatmap'. SNPs with an $r^2 \geq$ 0.5 were considered in strong LD.

## Genetic mapping of *Stb9*

A genetic linkage map was available for the Beaver/Soissons DH population and contains 312 polymorphic loci: 121 SSR, 49 AFLP, 135 DArT markers and 7 major genes [46]. The map covers 1,526 cM with an average density of one marker per 12.6 cM. QTL mapping was conducted using RAUDPCs and RAUDPCn phenotypes obtained from the isolate IPO09593 (*AvrStb9*) with R/qtl software v.1.48–1 [56]. Linkage analysis was performed for each trait with an initial Simple Interval Mapping (SIM), followed by a Composite Interval Mapping (CIM). For SIM, 1000 genome-wide permutations were used to calculate the significant logarithm of odds (LOD) threshold. Only QTLs that showed *p*-values less than 0.05 were considered significant. Association mapping in the BW panel was done following the exact same procedure as in the fungal population (see GWAS analyses). The 220 BW varieties have been previously genotyped

on an Affymetrix Axiom 410K single nucleotide polymorphism (SNP) array [57]. Procedure of allele calling, filtering and quality check of genotype data are described in Balfourier *et al.* [58]. For GWAS analysis, we excluded markers with less than 5% minor allele frequency and more than 10% missing data and kept a set of 186,961 SNPs that passed quality control criteria and could be anchored on the IWGSC annotation v2.1 [59]. We used filtered SNP data and phenotypes (RAUDPCs and RAUDPCn) from the isolates IPO323 (*avrStb9*) and IPO09593 (*AvrStb9*) to perform marker-trait associations.

## Gene annotation and expression analysis

Predicted genes in the candidate region were retrieved from the *Z. tritici* reference genome annotation [60]. We used RNA-seq data [23] mapping onto the reference genome IPO323 to manually curate *Zt_1_693* (*AvrStb9)* gene model. Canonical and non-canonical splicing sites were confirmed by *de-novo* transcriptome assembly obtained with DRAP [61]. The final gene model differed from those from previous annotations due to errors in predicting correctly introns and includes two missing exons. Candidate genes were screened for the presence of a secretion signal using SignalP 3.0 server [62]. Blastp was used to search for *AvrStb9* orthologous and paralogous proteins against the reference strain IPO323 and closely-related *Zymoseptoria* species. Protein domains were searched using Interproscan and structure homology prediction was performed in the HHpred server; <https://toolkit.tuebingen.mpg.de/tools/hhpred>. For gene expression analysis, RNA-seq data samples from *in vitro* and *in planta* (1, 4, 9, 14 and 21 dpi) conditions were obtained from Rudd *et al.* [23], and gene expression quantified as fragments per kilobase of exon per million fragments mapped (FPKM) using Cuffdiff v.2.2.1 [63]. *AvrStb9* DNA sequences are deposited in NCBI Genbank (accession numbers OP376566 and OP376567).

## Prediction and comparison of protein structures

The virulent avrStb9 and avirulent AvrStb9 (from isolates IPO323 and IPO09593, respectively) mature protein structures (i.e. without the signal peptide) were predicted with the ColabFold v1.5.2: AlphaFold2 using MMseqs2 with default parameters [64,65]. Structural similarity searches against the full Protein Data Bank (PDB) [66] were performed using the DALI webserver available at <http://ekhidna2.biocenter.helsinki.fi/dali/> [67]. Sequences and structures of the CPAF homodimer were extracted from the PDB with accession numbers 3DOR-A and 3DOR-B [25]. To further investigate similarities between protein tertiary structures, the RCSB PDB website <https://www.rcsb.org/alignment> [68] was used to align AvrStb9 against avrStb9 and AvrStb9 against CPAF (chains A and B) using the TM-align algorithm [69]. All images of 3D structures and structure superposition were captured and analyzed using Mol* Viewer <https://www.rcsb.org/3d-view> [70]. Finally, the five residues possibly associated with virulence were highlighted on the protein structures and distances between residues on both avrStb9 and AvrStb9 proteins were calculated using the measurement option of Mol* Viewer.

## Construction of disruption plasmids

For *AvrStb9* disruption, the hygromycin resistance marker (*hph*) was cloned between the 1,000 bp up- and downstream flanking regions of the ORF, respectively. The following DNA fragments were generated and cloned into pCAMBIA0380 plasmids (GenBank: AF234290.1). The up- and downstream flanks were obtained by PCR on gDNA of IPO323 with primer pairs 1&2 and 5&6 (S7 Table) respectively using Q5 Hot Start High fidelity DNA polymerase (NEB, Evry, France). The *hph* was amplified from pCAMBIA0380 with the primers 3 and 4 (S7 Table). pCAMBIA0380 was digested by *Eco*RI and *Xho*I and used for the cloning of three PCR

fragments with Gibson Assembly cloning kit (NEB, Evry, France) according to the supplier's indications. The ligation product was introduced into NEB 5α competent *Escherichia coli* cells (NEB, Evry, France), transformants selected on kanamycin (50 μg.mL$^{-1}$) containing LB-broth. Colonies were screened by PCR with the primer-pairs used for cloning. Four positive colonies were chosen for plasmid extraction (Macherey & Nagel, Düren, Germany) and restriction analysis. The construct was then introduced by electroporation into the *Agrobacterium tumefaciens* Agl1 strain. Transformants were selected and isolated on YEB (5 g. L$^{-1}$ beef extract, 1 g. L$^{-1}$ yeast extract, 5 g. L$^{-1}$ peptone, 5 g. L$^{-1}$ sucrose, 0.5 g. L$^{-1}$ MgCl2, 1.5% agar) with kanamycin (50 μg. mL$^{-1}$).

## Gene replacement constructs

To swap *AvrStb9* alleles between virulent (IPO323) and avirulent (IPO09593) strains, the following replacement cassettes were constructed. The respective *AvrStb9* alleles were amplified with primers 11 and 12 (S7 Table) from IPO323 or IPO09593 gDNA. These fragments cover the ORF with 1 kb upstream of the START codon and 360 bp downstream of the STOP codon. The *hph* (amplified with primers 9 and 10 from pCAMBIA0380 plasmid DNA) and the 1 kb upstream flank of *AvrStb9* gene (amplified with primers 7 and 8) were ligated to *AvrStb9* alleles into the *EcoRI-XhoI* digested pCAMBIA0380, with Gibson Assembly cloning kit (NEB, Evry, France). Plasmids were introduced into NEB 5α competent *E. coli* cells (NEB, Evry, France) and transformants selected on kanamycine (50 μg.mL$^{-1}$) containing LB-broth. Colonies were screened by PCR with the primer-pairs used for cloning. Four positive colonies were chosen for plasmid extraction (Macherey & Nagel, Düren, Germany) and sequence analysis. These constructs named respectively pR323 and pR9593 were then introduced by electroporation into the *Agrobacterium tumefaciens* Agl1 strain. Transformants were selected and isolated on YEB with kanamycin (50 μg.mL$^{-1}$).

## Generation of *Z. tritici* transformants

The transformation procedure was adapted from Zwiers & de Waard [71] and was performed as previously described [72]. For *AvrStb9* deletion, the following *Z. tritici* strains were transformed with the disruption plasmid: IPO323, IPO09593, IPO09359, IPO09139 and SYN32. Transformants were selected and isolated twice on hygromycin (100 μg. mL$^{-1}$) containing medium. The replacement of the *AvrStb9* ORF by the *hph* gene was checked by PCR on gDNA with the primer pair 13 and 14. The gene deletion is expected to give a 1.6 kb fragment instead of the native 2.4 kb amplicon. The total number of isolated and analyzed transformants were: IPO323 (n = 144), IPO09593 (n = 23), IPO09359 (n = 26), IPO09139 (n = 46), SYN32 (n = 38). One single transformant produced the expected PCR profile, namely SYN32_ΔT12.

For *AvrStb9* allele swaps, IPO323 was transformed with plasmids pR323 and pR9593. Transformants selected and isolated twice on hygromycin containing medium (100 μg. mL$^{-1}$). The replacement of *AvrStb9* alleles in the transformants was checked by PCR using primers 15 and 16, leading to a single 1.6 kb amplicon in replacement mutants and to an additional 238 bp amplicon in ectopic transformants. Transformants with the expected single band size were PCR checked with primers 17 and 18 as these primers allow to distinguish the deletion of the amino acid residues 66–71 in the IPO09593 background from the IPO323 one, and subsequently submitted to sequence analysis with primers indicated in S7 Table.

## Phylogenetic analysis

We used OrthoMCL algorithm to search for *AvrStb9* orthologous and paralogous protein sequences using a cutoff E-value less than 1e-20 and filtering for low complexity regions [73];

<https://orthomcl.org/orthomcl/app/>. Of the 148 putative *AvrStb9* orthologs/paralogs identified through OrthoMCL, we selected 122 by removing redundant information using the following criteria: (*i*) when multiple identical sequences were present, one was arbitrarily chosen and retained; and (*ii*) when multiple sequences from different strains of the same species were present, an arbitrarily chosen strain (the reference strain, if known) was retained. A multiple sequence alignment was then generated with Clustal Omega v1.2.4 [74] and alignment cleaning was performed using trimAL with the—gappyout option [75]. We constructed a maximum likelihood tree of selected *AvrStb9* orthologs/paralogs using RaxML with the GAMMA JTT model and 100 bootstrap replicates [76]. Phylogenetic trees were visualized using iTOL [77]; <https://itol.embl.de/>.

## Population genetics and selection analyses

In addition to the fungal population used for genetic analyses, we extended our sequence diversity and selection analyses to include a global *Z. tritici* population originating from four distinct geographic locations, including the middle East (n = 30, Israel), Australia (n = 27, Wagga Wagga), Europe (n = 33, Switzerland) and North America (n = 56, Oregon, USA). Procedures of *de novo* genome assemblies and sequence extraction were previously described [78]. We used DnaSP v6 [79] to calculate summary statistics of the population genetic parameters associated with *AvrStb9*. Sliding window analyses of nucleotide diversity ($\pi$) and Tajima's D were conducted in a window length of 200 bp and a step size of 20 bp. To test the selective pressure acting on *AvrStb9*, we estimated the rates of non-synonymous (dN) and synonymous (dS) substitutions using CodeML from the PAML package [80]. A dN/dS ($\omega$) ratio of 1 indicated neutrality, whereas $\omega < 1$ or $> 1$ suggested purifying or diversifying selection, respectively. However, the presence of intragenic recombination could bias the $\omega$ rate testing. To account for these issues, we performed the analysis with a non-recombining phylogenetic tree generated from the RDP4 software using RaxML with 100 bootstraps [81].

## Supporting information

**S1 Table. Mean disease phenotypes measured among wild-types, mutant and chimeric strains of AvrStb9 alleles.**
(XLSX)

**S2 Table. Summary statistics of GWA analysis on Stb9 resistance in the BW panel.** Only SNPs above the Bonferroni-corrected threshold are presented.
(XLSX)

**S3 Table. List of the top 10 protein tertiary structures with the highest similarity to AvrStb9 protein structure (from DALI webserver; accessed in january 2023).**
(XLSX)

**S4 Table. List of the pairwise structure alignment results.**
(XLSX)

**S5 Table. List of AvrStb9 selected orthologs/paralogs identified by blastp search using OrthoMCL database.**
(XLSX)

**S6 Table. Model test and parameter estimates of diversifying selection with codeml based on the total *AvrStb9* data set.**
(XLSX)

**S7 Table. List of primers used in this study.**
(XLSX)

**S1 Fig. Density distribution of percentage of leaf area covered by pycnidia (PLACP) measured at 21 days post inoculation (dpi) on the wheat cultivar 'Soissons' in the *Z. tritici* fungal population.**
(PDF)

**S2 Fig. Population structure and relatedness in the fungal population used for GWAS analysis.** (a) Principal component analysis using full SNP data. PC1, PC2 and PC3 explained 1.30%, 1.12% an 0.69% of the phenotypic variation, respectively. (b) Heatmap of the kinship matrix using VanRaden method [55].
(PDF)

**S3 Fig. Manhattan plots of PLACN (top) and PLACP (bottom) on the wheat cultivar 'Soissons'.** The horizontal line indicates the genome-wide significance threshold (Bonferroni correction at α<0.05).
(PDF)

**S4 Fig. Improved *AvrStb9* gene model using RNA-seq reads at 9 dpi mapped to the reference genome IPO323 [45,60].**
(PDF)

**S5 Fig. PLACP of wild-type, mutant, chimeric strains and the single knock-out mutant phenotyped on cultivars carrying or not the resistance gene *Stb9* and the susceptible cultivar 'Taichung-29'.**
(PDF)

**S6 Fig. Distribution of the relative AUDPC sporulating leaf (RAUDPCs) area in the panel of bread wheat cultivars (a) and the lead SNP of *Stb9* locus mapped using GWAS in the same panel (b).**
(PDF)

**S7 Fig. *AvrStb9* homologs and their expression profile.** (a) A phylogenetic tree of *AvrStb9* paralogs and orthologs from *Z. tritici* and its relative species (*Zt = Z.tritici; Zpse = Z. pseudotritici; Zbre = Z.brevis; Zard = Z. ardabilae*). (b) Gene expression profile of *AvrStb9* paralogs using IPO323 RNAseq data [23].
(PDF)

**S8 Fig. Disease scale used for the visual assessment of percent of leaf area covered by necrosis (PLACN) and by pycnidia (PLACP) after inoculation of wheat leaves with the phytopathogenic fungus *Zymoseptoria tritici*.**
(PDF)

## Acknowledgments

We thank David Gouache, Henriette Goyeau, Gert H.J. Kema, Marc-Henri Lebrun, Frédéric Suffert and Anne-Sophie Walker for their contribution in establishing the fungal population; Johann Confais for his help in the virulence assays; Robert King and Jason Rudd for providing IPO323 RNA-seq data; Florence Cambon, the VégéPôle and GENTYANE platforms for the development of the Courtot NILs; Marc-Henri Lebrun for critical reading of the manuscript and valuable feedback; and Sophie Bouchet and Fanny Hartmann for assistance with the initial GWAS analyses.

## Author Contributions

**Conceptualization:** Sabine Fillinger, Thierry C. Marcel.

**Data curation:** Nicolas Lapalu.

**Formal analysis:** Reda Amezrou, Jérôme Compain, Nicolas Lapalu, Clémentine Louet, Daniel Croll, Sabine Fillinger, Thierry C. Marcel.

**Funding acquisition:** Joëlle Amselem, Sabine Fillinger, Thierry C. Marcel.

**Investigation:** Reda Amezrou, Colette Audéon, Sandrine Gélisse, Aurélie Ducasse, Clémentine Louet, Sabine Fillinger, Thierry C. Marcel.

**Project administration:** Joëlle Amselem, Thierry C. Marcel.

**Resources:** Cyrille Saintenac, Simon Orford, Daniel Croll, Thierry C. Marcel.

**Supervision:** Joëlle Amselem, Sabine Fillinger, Thierry C. Marcel.

**Validation:** Reda Amezrou, Sabine Fillinger, Thierry C. Marcel.

**Visualization:** Reda Amezrou, Clémentine Louet, Thierry C. Marcel.

**Writing – original draft:** Reda Amezrou.

**Writing – review & editing:** Reda Amezrou, Cyrille Saintenac, Nicolas Lapalu, Clémentine Louet, Daniel Croll, Sabine Fillinger, Thierry C. Marcel.

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
