## [Decision Letter · Decision Letter 0]

27 Dec 2022

Dear Dr. MARCEL,

Thank you very much for submitting your manuscript "A secreted protease-like protein in Zymoseptoria tritici is responsible for avirulence on Stb9 resistance gene in wheat" for consideration at PLOS Pathogens. As with all papers reviewed by the journal, your manuscript was reviewed by members of the editorial board and by several independent reviewers. The reviewers appreciated the attention to an important topic. Based on the reviews, we are likely to accept this manuscript for publication, providing that you modify the manuscript according to the review recommendations.

Each of the three reviewers appreciate the important findings reported in the manuscript. However, several important points are raised, notably with respect to the presentation of results and discussion of the findings. Hereby, one reviewer mentions the outdated notion of effectors being small secreted proteins without functional domains; a definition which the authors repeatedly put forward in the manuscript.

Another important point is the PLACN data which was used in the GWAS analyses to identify AvrStb9. The authors should provide leaf images to illustrate the typical symptoms on the wheat cultivars used in the study. Moreover, it is confusing that PLACN was the phenotypic parameter used for the GWAS, but subsequent analyses were based on PLACP, pycnidia density. The authors should clarify this point and ideally provide both pycnida and necrosis data from their experiments.

The authors should also include structural predictions of AvrStb9, as suggested by one reviewer. This will allow the overlay with other known proteases and provide further evidence to hypotheses regarding protein functions. Moreover, the authors can position the polymorphic sites on the predicted structure to develop further hypotheses regarding functional relevance of the variable sites.

In the revised manuscript, the authors should carefully consider each point raised by the three reviewers.

Sincerely,

Eva H. Stukenbrock, PhD

Academic Editor

PLOS Pathogens

Bart Thomma

Section Editor

PLOS Pathogens

Kasturi Haldar

Editor-in-Chief

PLOS Pathogens

orcid.org/0000-0001-5065-158X

Michael Malim

Editor-in-Chief

PLOS Pathogens

orcid.org/0000-0002-7699-2064

Each of the three reviewers appreciate the important findings reported in the manuscript. However, several important points are raised, notably with respect to the presentation of results and discussion of the findings. Hereby, one reviewer mentions the outdated notion of effectors being small secreted proteins without functional domains; a definition which the authors repeatedly put forward in the manuscript.

Another important point is the PLACN data which was used in the GWAS analyses to identify AvrStb9. The authors should provide leaf images to illustrate the typical symptoms on the wheat cultivars used in the study. Moreover, it is confusing that PLACN was the phenotypic parameter used for the GWAS, but subsequent analyses were based on PLACP, pycnidia density. The authors should clarify this point and ideally provide both pycnida and necrosis data from their experiments.

The authors should also include structural predictions of AvrStb9, as suggested by one reviewer. This will allow the overlay with other known proteases and provide further evidence to hypotheses regarding protein functions. Moreover, the authors can position the polymorphic sites on the predicted structure to develop further hypotheses regarding functional relevance of the variable sites.

In the revised manuscript, the authors should carefully consider each point raised by the three reviewers.

Reviewer Comments (if any, and for reference):

Reviewer's Responses to Questions

**Part I - Summary**

Reviewer #1: This manuscript describes the identification of AvrStb9 from Z. tritici by GWAS. For this the authors re-sequenced 103 isolates and the data were made available on NCBI. The GWAS approach lead to the identification of a specific location, but the genetic confirmation of the secreted protein to be AvrStb9 appeared hindered by the inability of generating AvrStb9 knock out lines suggesting that knock out of this conserved gene may be lethal. The dominant effect of an avirulence effector over virulent variants however allowed confirmation of the gene-for-gene relationship underling STB9 resistance mediated by the here identified AvrStb9.

Although there is little functional data provided, I consider this work as well suited for Plos Pathogens, in particular with respect to the few Avr effectors (or effectors in general) characterized for this wheat pathogen.

I have some minor comments that should improve accessibility of the data.

Reviewer #2: The manuscript by x et al., describes the genetic mapping, population level polymorphism, expression and subsequent validation of a novel fungal avirulence gene from the globally important wheat fungal pathogen Zymoseptoria tritici.

The authors used quantitative disease assays, whole genome sequences and corroborating in planta gene expression data to identify the specific polymorphisms in a putative S41 secreted protease as being responsible for recognition (or lack of) by the STB disease resistance QTL / gene Stb9. This R gene has yet to be cloned although this study also improved the resolution of its mapping and conclusively demonstrated the gene-for-gene interaction. The quality of the mapping and candidate gene selection approach is excellent, and by performing allele swaps the authors demonstrated that the protease was indeed AvrStb9. Unfortunately, efforts to generate knock-out mutants provided evidence for AvrStb9 being a potential essential gene (in itself interesting), but this precluded its formal testing for a role in virulence on susceptible cultivars. However, wide population level analysis suggested that the protein is always present and functional whilst being under positive selection. Overall, this is strong work and the manuscript is very nicely written and presented.

My only concern is whether there are sufficient mechanistic insights presented. It is a pity that further characterisation of the putative avr protease was not performed. The authors themselves raise many interesting testable further questions. In particular whether protease activity was required for avr function? Or whether a small region (peptide) might confer the recognition? Is it truly an essential gene? On the other hand, if the effector had been a small cysteine rich protein of unknown function, these questions would not be asked. On that basis I can understand the authors wish to quickly release this interesting information on a novel Avr and save detailed functional characterisation until a later date. I must therefor leave it to editorial judgement on whether the manuscript provides enough mechanistic information in its current form. I can say that I believe the novel avr identity alone will draw many citations.

Reviewer #3: The authors describe the identification of a secreted protease from Zymoseptoria tritici that is likely AvrStb9, the protein responsible for avirulence on wheat varieties containing the Stb9 resistance gene. This work is important as this is only the third described Avr gene from this widespread crop pathogen. The authors use a combination of techniques including genetic mapping and reverse genetics to show the likely identity of AvrStb9, and also the corresponding Stb9 gene in wheat. Further, they make use of diversity present at the AvrStb9 locus in the pathogen population to make predictions on specific residues in AvrStb9 that are important for virulence.

Overall I found this paper to be interesting, comprehensive and well-presented. There is a good level of detail used when describing how experiments were performed and I believe these experiments to have been performed with a high level of competency. The subject matter is of great value both to the Zymoseptoria research community and the wider plant-microbe community given that this class of protease effector appear conserved across many species. I have some specific comments detailed below related to the presentation of some data that could help improve the manuscript:

(1) For Figure 2, it would be helpful to have leaf images showing typical symptoms of Soissons and the Courtot NILs with the different allele swap strains. It would also be helpful to see PLACN data either as part of this figure or as supplemental figure in addition to the PLACP data that is already presented. This is important as GWAS in Fig.1 found the AvrStb9 candidate locus significance based on PLACN but not PLACP. There needs to be some additional explanation to why the GWAS found the locus based on PLACN, but the authors have used PLACP as the metric for scoring disease symptoms in subsequent experiments.

(2) AvrStb9 structure – The section on structural biology is somewhat superficial at present. It would be helpful to show a structural prediction of AvrStb9 protease (AlphaFold, EBI), with comparison or overlay against the S41 protease class (if experimentally validated structures are available). Highlighting of key residues for escaping Stb9 recognition could be highlighted on this model. This would illustrate whether these key residues are separate from putative protease function. I would assume they are separate given the difficulty in obtaining a gene KO of this protease gene (as authors state). If mutations are found in protease domain, this needs to be discussed - does this not impact protease function at all, or could it impact substrate specificity. A model could also highlight active site residues, a brief literature search suggests this is a catalytic tetrad that has been partly characterized, so this should be easy to illustrate.

**Part II – Major Issues: Key Experiments Required for Acceptance**

Reviewer #1: One thing I consider as major issue in the current version of the manuscript is the insistence of the authors (abstract, introduction and discussion) that AVR effectors are generally considered small secreted proteins without functional domains. This is a very outdated and narrow view and should as such not be present in a publication with relatively broad readership.

Example: Line 40 cont: It is commonly assumed that most Avr genes encode small secreted proteins, with no predicted functional domain and no sequence homology in other species. But the use of methods without a priori on the function or structure of genetic determinants may allow to identify Avr genes coding for proteins not falling into this category.

- I assume the authors try to differentiate their here generated data (AvrStb9 is rather large and carries predicted protease domains) from previous reports by using the statement as selling point. However, this is simply not correct. First, only fungal effectors are frequently referred to as small secreted proteins, but even for fungi, not all characterized effectors are small. Hundreds of Avr genes have been predicted to carry functional domains. The best studies ones in this sense are likely bacterial Avrs, but also fungal Avrs were shown to have functional enzymatic activity (eg. Z. tritici AvrStb6). Similarly, the discussion associated does not give any deep insight and is rather a selective list of examples.

-Considering the limited knowledge on functional studies associated with (avirulence) effectors provided here, I suggest to not go further into this topic or to comment on effector functions/domain functions. It will be sufficient to mention that unlike many other effectors, this one here appears large in size. (Many effectors actually appear to have predicted functional domains, especially when looking at their predicted structures. However, as for here-identified AvrStb9, it was not (yet) clarified weather these domains are functional. (This is generally often ignored). For this reason, I also suggest to remove the reference to ‘protease-like’ from the title.)

Reviewer #2: (No Response)

Reviewer #3: (No Response)

**Part III – Minor Issues: Editorial and Data Presentation Modifications**

Reviewer #1: Clarity: For generating easy data access to the readership, I suggest the authors to indicate in the graphs (Fig 2) if an isolate is known to be avirulent or virulent on Stb9 (eg….. IPO323 (Stb6_vir) ….).

Fig. 2 a IPO09593 appears generally weakly virulent also in -stb9 lines. With the statistics applied here (I assume it is an anova test) the data suggests no statistical difference between the virulence of the isolate between +Stb9 and -Stb9 lines. This may confuse the readership but is likely simply an artefact because of the way the stats are applied. A non-parametric test (for example Kruskal-Wallis followed by Dunn’s test) is more appropriate here (either way, the test applied should be indicated in the legend). It may also be useful to separately analyses the differences for each wheat background (if so, please also indicate in the legend).

Clarity: Define PLACP in figure legends and maybe indicate y axis as ‘necrosis (PLACP)’

Line 51: please change to ‘for SOME OF the most….’

Line 53: play critical roleS

Line 107: are avirulent on ‘soissons’

Discussion: The Avr gene is expressed at the timepoint of lifestyle transitioning from bio to necrotrophy. At this point of our knowledge, this is rather rare. Only few Avrs have been characterized for hemi-biotrophs, but the best studies ones (eg. from M. oryzae) are recognized during the biotrophic phase. This cuts off infection early on. And in agreement these effectors induce R-gene dependent cell death in heterologous overexpression assays.

The late recognition of AvrStb9 could suggest an host STB9 resistance function that inhibits transition to necrotrophy, i.e. inhibition of cell death. This is purely speculative at this stage as heterologous assays are not shown (and I am not asking the authors to perform these for this manuscript), but it may be an interesting point to discuss in the manuscript so that assays testing such hypothesis may be considered in the future. This point could replace the statements about effectors generally being SSP.

Reviewer #2: -It is stated several times that recognition triggers “a strong immune response” but no responses are actually measured, only lack of disease symptoms. How do you know that a low or moderate immune response might not already be sufficient to prevent the disease? Suggest rephrase.

-Whilst detailed functional analysis of the putative protease is planned for the future it would have been nice to view the polymorphic residues in the context of the positions of essential amino acids likely required for catalytic activity (eg using NCBI conserved domain databases or similar) ie it would be nice the see the position of polymorphic amino acids in greater resolution than the current Fig 3C. Similarly, it would be nice to run alphafold, or a structural prediction software, to see if the polymorphism likely affects folding or overall protein structure or alters accessibility to the catalytic domain.

-Given that the journal has a broad readership (including many studying disease of animals etc) I think it would be useful to include at least a few photos of infected leaf tissues so that a non-specialist reader can make sense of the quantitative data such as that shown in Fig2. Maybe in supplementary if not in main? Within the Zymoseptoria community we are aware of how these things look, but it is certainly not true for researchers from disciplines further afield.

-Line 345 discussion highlights some potential means by which avrStb9 and Stb9 might interact but does not consider the “Guard” model or that the protease might produce a peptide which is recognised by Stb9???

-The authors are correct in stating that large enzymes are not typical for fungal avr’s but they are quite well known to be recognised by receptor-like proteins during PTI (for examples see Plant Cell. (2004) 16: 1604–1615; EMBO J (2002) 21(24):6681-8). Indeed, there is now a well-established blurring between ETI and PTI (Plant Cell (2011) 23(1):4-15). PAMP’s are also often essential and cannot be lost or mutated (alike AvrStb9???). It would be nice to perhaps have a little more discussion on the possibility that Zymoseptoria might co-opt more typical PTI mechanisms to establish ETI?

Reviewer #3: I found several presentational errors related to figures and their associated legends;

Label for Figure 1B,D is out of place on figure. Legend confuses parts B and C.

Figure 2 has no part B, labelled only as parts A, C and D.

Figure 3 sections B and C don't match with legend.

PLOS authors have the option to publish the peer review history of their article (what does this mean?). If published, this will include your full peer review and any attached files.

Reviewer #1: No

Reviewer #2: No

Reviewer #3: **Yes: **Graeme J Kettles

Figure Files:

Data Requirements:

Reproducibility:

References:

---

## [Decision Letter · Decision Letter 1]

19 Apr 2023

Dear Dr. MARCEL,

We are pleased to inform you that your manuscript 'A secreted protease-like protein in Zymoseptoria tritici is responsible for avirulence on Stb9 resistance gene in wheat' has been provisionally accepted for publication in PLOS Pathogens.

Best regards,

Eva H. Stukenbrock, PhD

Academic Editor

PLOS Pathogens

Bart Thomma

Section Editor

PLOS Pathogens

Kasturi Haldar

Editor-in-Chief

PLOS Pathogens

orcid.org/0000-0001-5065-158X

Michael Malim

Editor-in-Chief

PLOS Pathogens

orcid.org/0000-0002-7699-2064

The authors have addressed the comments raised by the three reviewers.

One point is still to which extent the author can term this new Avr a "protease". As no functional data is presented in the study (underlined by reviewer 2 in the first round of reviews), the authors should be careful with the claim. In the manuscript, the authors have now included more careful statements with respect to protease activity. However, a strong statement is made in the title by calling the Avr "protease-like". As pointed out by Reviewer 1, the authors should re-formulate the title to avoid any confusion regarding something which is only a "putative" function.

Reviewer Comments (if any, and for reference):

Reviewer's Responses to Questions

**Part I - Summary**

Reviewer #1: (No Response)

**Part II – Major Issues: Key Experiments Required for Acceptance**

Reviewer #1: (No Response)

**Part III – Minor Issues: Editorial and Data Presentation Modifications**

Reviewer #1: The authors have answered my previous questions and made the adjustments I considered important with one exception: I suggested to remove the reference to ‘protease-like’ effector from the title as the authors do not show any functional data and cannot conclude that the effector is a protease.

The authors argue that the term ‘protease-like’ is valid as based on homology. I agree on this, but it becomes more and more clear that sequence and structural homology may not relate to effector function, e.g. that whole effector families from different fungi show strong homologies to certain enzymes (e.g. 10.1038/s41564-022-01287-6) but that the effectors itself have neither enzymatic activity nor may this function be related to the effector’s virulence function. This is an important point that is just being investigated. Currently it is speculated that certain folds are preferable due to high protein stability, easy protein transport or other advantageous properties. This may not be the case here, but because there is no functional data supporting any proteolytic function for the identified Avr, it cannot be excluded. In order to not mislead the field and in particular early career scientists that may struggle with the interpretation of scientific results, I suggest to abstain from the reference to a protease in the title.

PLOS authors have the option to publish the peer review history of their article (what does this mean?). If published, this will include your full peer review and any attached files.

Reviewer #1: No

---

## [Editor Report · Acceptance letter]

9 May 2023

Dear Dr. MARCEL,

We are delighted to inform you that your manuscript, "A secreted protease-like protein in Zymoseptoria tritici is responsible for avirulence on Stb9 resistance gene in wheat," has been formally accepted for publication in PLOS Pathogens.

Best regards,

Kasturi Haldar

Editor-in-Chief

PLOS Pathogens

orcid.org/0000-0001-5065-158X

Michael Malim

Editor-in-Chief

PLOS Pathogens

orcid.org/0000-0002-7699-2064